# D-Aspartate Depletion Perturbs Steroidogenesis and Spermatogenesis in Mice

**DOI:** 10.3390/biom13040621

**Published:** 2023-03-30

**Authors:** Alessandra Santillo, Sara Falvo, Massimo Venditti, Anna Di Maio, Gabriella Chieffi Baccari, Francesco Errico, Alessandro Usiello, Sergio Minucci, Maria Maddalena Di Fiore

**Affiliations:** 1Dipartimento di Scienze e Tecnologie Ambientali, Biologiche e Farmaceutiche, Università degli Studi della Campania ‘Luigi Vanvitelli’, 81100 Caserta, Italy; 2Dipartimento di Medicina Sperimentale, Sez. Fisiologia Umana e Funzioni Biologiche Integrate, Università degli Studi della Campania ‘Luigi Vanvitelli’, 80138 Napoli, Italy; 3Laboratorio di Neuroscienze Traslazionali, CEINGE Biotecnologie Avanzate, Franco Salvatore, 80145 Napoli, Italy; 4Dipartimento di Agraria, Università degli Studi di Napoli “Federico II”, 80055 Portici, Italy

**Keywords:** steroidogenesis, spermatogenesis, D-aspartate oxidase, D-aspartate, DAAM1, PREP, knockin mice

## Abstract

High levels of free D-aspartate (D-Asp) are present in vertebrate testis during post-natal development, coinciding with the onset of testosterone production, which suggests that this atypical amino acid might participate in the regulation of hormone biosynthesis. To elucidate the unknown role of D-Asp on testicular function, we investigated steroidogenesis and spermatogenesis in a one-month-old knockin mouse model with the constitutive depletion of D-Asp levels due to the targeted overexpression of D-aspartate oxidase (DDO), which catalyzes the deaminative oxidation of D-Asp to generate the corresponding α-keto acid, oxaloacetate, hydrogen peroxide, and ammonium ions. In the *Ddo* knockin mice, we found a dramatic reduction in testicular D-Asp levels, accompanied by a significant decrease in the serum testosterone levels and testicular 17β-HSD, the enzyme involved in testosterone biosynthesis. Additionally, in the testes of these *Ddo* knockin mice, the expression of PCNA and SYCP3 proteins decreased, suggesting alterations in spermatogenesis-related processes, as well as an increase in the cytosolic cytochrome c protein levels and TUNEL-positive cell number, which indicate an increase in apoptosis. To further investigate the histological and morphometric testicular alterations in *Ddo* knockin mice, we analyzed the expression and localization of prolyl endopeptidase (PREP) and disheveled-associated activator of morphogenesis 1 (DAAM1), two proteins involved in cytoskeletal organization. Our results showed that the testicular levels of DAAM1 and PREP in *Ddo* knockin mice were different from those in wild-type animals, suggesting that the deficiency of D-Asp is associated with overall cytoskeletal disorganization. Our findings confirmed that physiological D-Asp influences testosterone biosynthesis and plays a crucial role in germ cell proliferation and differentiation, which are required for successful reproduction.

## 1. Introduction

Steroidogenesis and spermatogenesis are regulated through a complex interaction between hormones and intracellular biochemical signaling pathways. Testis steroidogenesis is modulated by various factors, among which, D-aspartate (D-Asp) has received much attention [1,2,3,4,5,6]. Specifically, D-Asp is present in the Leydig cells (LC), Sertoli cells (SC), and germ cells (GC), notably in spermatogonia (SPG), elongate spermatids (SPT) [7] and spermatozoa (SPZ) [8,9], of rodent and human testes. The levels of D-Asp and testosterone were found to be strongly correlated in rat testes, as the synthesis of both molecules starts at the late fetal stage and progressively increases until sexual maturity [10].

The peroxisomal enzyme, D-aspartate oxidase (DDO, D-AspO; EC1.4.3.1), specifically metabolizes D-Asp into α-oxaloacetate, ammonia, and hydrogen peroxide. The reciprocal localization of DDO and D-Asp suggests that this enzyme depletes endogenous stores of D-Asp [11].

Several studies have shown that intraperitoneal and oral administration of D-Asp to adult rats resulted in its accumulation in the testis and induced an increase in the levels of serum luteinizing hormone, progesterone, and testis/serum testosterone [12,13].

Studies have shown that D-Asp participates in the synthesis of dihydrotestosterone and 17β-Estradiol by activating 5α-Reductase and P450 aromatase, respectively [14,15,16]. Hence, D-Asp can induce testosterone synthesis either through the hypothalamus–pituitary–testis axis or by directly acting on LC [12,17,18]. The evidence for the action of D-Asp via the second pathway emerged from several in vitro studies [17,18,19,20,21,22,23]. Specifically, in rat LC, D-Asp, alone or in the presence of human chorionic gonadotropin, upregulated testosterone synthesis by stimulating the gene and protein expression of the steroidogenic acute regulatory protein (StAR), which is a transport protein that regulates cholesterol transfer within the mitochondria [17,18,19,20]. Other studies conducted on murine SPG (GC-1) and spermatocyte (SPC) (GC-2) cell lines have also shown a direct role of D-Asp in spermatogonial and spermatocytic proliferation, respectively [21,22,23].

Cell differentiation during spermatogenesis requires a proper cytoskeleton dynamic, regulated by several proteins [24], such as prolyl endopeptidase (PREP) and disheveled-associated activator of morphogenesis 1 (DAAM1), which play key roles in the postnatal development of the rat testis [25,26] and, as for PREP, in sperm motility [27,28]. Moreover, as we have found their altered expression and localization in pathological and/or altered spermatogenesis [29,30,31,32], we raised the possibility of their use as potential new markers of proper fertility. Specifically, PREP is a binding partner of tubulin [25]; it also directly limits GnRH stimulation of the pituitary gonadotropes by cleaving its C-terminal glycinamide residue, as demonstrated in rat and ovine hypothalamic extracts [33,34]. DAAM1 belongs to the formin family and promotes actin polymerization [26]. In previous studies, we showed that D-Asp modulates the expression of PREP and DAAM1 in the rat testis [35,36].

To elucidate the role of D-Asp in steroidogenesis and spermatogenesis, we used a knockin mouse model (*R26^Ddo^*^/*Ddo*^) in which the endogenous D-Asp levels were constitutively depleted through the overexpression of DDO from the earliest embryonic stages of development [37].

## 2. Materials and Methods

### 2.1. Ddo Knockin Mice

The *Ddo* knockin mouse model was generated through the recombinant targeting of an inducible *Ddo*-*LacZ* cassette into the genomic locus of *Rosa26* (*R26*), thus enabling the ectopic expression of *Ddo* under the regulatory control of the constitutive *R26* promoter and, in turn, the constitutive depletion of D-Asp starting from the very early stages of embryonic development [37]. Heterozygous *Ddo* knockin mice (*R26^+^*^/*Ddo*^) were interbred to generate wild-type (*Ddo^+^*^/*+*^) and homozygous (*R26^Ddo^*^/*Ddo*^) knockin mice, which were used for all the experiments. Mice were genotyped by PCR according to the protocol previously reported [37]. Animals were housed in groups (4–5 per cage), at a constant temperature (22 ± 1 °C) on a 12 h light/dark cycle (lights on at 7 AM) with food and water ad libitum. All research involving animals was carried out in accordance with the directive of the Italian Ministry of Health governing animal welfare and protection (D.LGS 26/2014) and approved by “Direzione Generale della Sanità e dei Farmaci Veterinari (Ufficio 6)” (permission nr 796/2018).

### 2.2. High-Performance Liquid Chromatography Analysis

D-Asp and L-Asp levels in mouse testis samples were analyzed using HPLC [38,39]. Briefly, samples were homogenized in 1:10 (*w*/*v*) 0.2 M TCA, sonicated (3 cycles, 10 s each), and centrifuged at 13,000× *g* for 20 min. The precipitated protein pellets were stored at −80 °C for protein quantification. The total protein content of homogenates was determined by using the Bradford assay method, after resolubilizing the TCA-precipitated protein pellets. The supernatants were neutralized with NaOH and subjected to precolumn derivatization with o-phthaldialdehyde/N-acetyl-L-cysteine. Diastereoisomer derivatives were resolved on a Simmetry C8 5-μm reversed-phase column (Waters, 4.6 × 250 mm). The total concentration of amino acids detected in tissue homogenates was normalized by the total protein content and expressed as nmol/mg protein. Identification and quantification of free D-Asp and L-Asp levels were based on retention times and peak areas, compared with those associated with external standards. The identity of peaks was confirmed by adding known amounts of external standards. The identity of the D-Asp peak was also evaluated by the selective degradation catalyzed by a recombinant human DDO (hDDO) [40]. The hDDO enzyme (12.5 μg) was added to the samples, incubated at 30 °C for 3 h, and subsequently derivatized.

### 2.3. Testosterone Assays

Testosterone levels were determined in both serum and testis of Ddo knockin and wt mice using a testosterone enzyme immunoassay kit (DiaMetra, Milan, Italy) [41,42]. Briefly, testes were homogenized 1:10 (*w*/*v*) with PBS 1×. The homogenate was then mixed with ethyl ether (1:10 *v*/*v*) and the ether phase was withdrawn after centrifugation at 3000× *g* for 10 min. The upper phase (ethyl ether) was transferred to a glass tube and was left to evaporate on a hot plate at 40 °C to 50 °C under a hood. The residue was dissolved in 0.25 mL of 0.05 M sodium phosphate buffer, pH 7.5, containing BSA at a concentration of 10 mg/mL, and then used for the assay. The sensitivities were 32 pg/mL for testosterone.

### 2.4. Histological and Morphometric Analyses

From each animal, the right testis was collected and fixed in 10% neutral buffered formalin for 48 h. After that, they were dehydrated through ascending grades series of ethanol, cleared in xylene, and finally, embedded in paraffin blocks. Five-micrometer slices were obtained, stained with hematoxylin and eosin (H&E), and examined under a light microscope [43]. For histological evaluation, 30 seminiferous tubules/animal (N = 8) for a total of 240 tubules per group were counted under a microscope (Leica DM 2500, Leica Microsystems, Wetzlar, Germany). Photographs were taken using the Leica DFC320 R2 digital camera. Histological parameters such as the diameter of the seminiferous tubules and the thickness of the germinal epithelium were measured with ImageJ software (National Institutes of Health, 156 Bethesda, MD, USA).

### 2.5. Immunofluorescence Analysis

For immunofluorescence (IF) staining, testis sections were processed according to Venditti et al. [36,44]. Briefly, the quenching of autofluorescence was performed by treating slides with 0.3 M glycine in phosphate-buffered saline (PBS) for 30 min, followed by a permeabilization step with PBS pH 7.4 containing 0.1% Triton-X-100 for 30 min. Later, sections were incubated overnight at 4 °C with primary antibodies described in Appendix A. After two washes in TPBS (PBS containing 0.25% Tween20) and two washes in PBS, the secondary antibodies diluted in the blocking mixture were added for 1 h at RT. Finally, slides were washed again, and the cells’ nuclei were marked with Vectashield + 4′,6-diamidino-2-phenylindole (DAPI; H-1200-10; Vector Laboratories, Peterborough, UK). The sections were observed and captured with an optical microscope (LeicaDM5000 B + CTR 5000) with a UV lamp and saved with IM 1000 software (version 4.7.0).

### 2.6. Protein Extraction and Western Blot Analysis

Each testis from *wild type* (*n* = 5) and *Ddo*/*Ddo* (*n* = 5) mice were homogenized directly in lysis buffer containing 50 mM HEPES, 150 mM NaCl, 1 mM EDTA, 1 mM EGTA, 10% glycerol, 1% Triton X-100 (1:2 *w*/*v*), 1 mM phenylmethylsulphonyl fluoride (PMSF), 1 g aprotinin, 0.5 mM sodium orthovanadate, and 20 mM sodium pyrophosphate, pH 7.4 (Sigma Chemical Corporation, St. Louis, MO, USA), then clarified by centrifugation at 14,000× *g* for 10 min. Protein concentration was determined by the Bradford assay (Bio-Rad, Melville, NY, USA). Fifty micrograms of total protein extracts were boiled in Laemmli buffer for 5 min before electrophoresis. The samples were subjected to SDS-PAGE (13% polyacrylamide) under reducing conditions. After electrophoresis, proteins were transferred onto a nitrocellulose membrane (Immobilon Millipore Corporation, Bedford, MA, USA). The complete transfer was assessed using prestained protein standards (Bio-Rad). The membranes were first treated for 1 h with blocking solution (5% non-fat powdered milk in 25 mM Tris, pH 7.4; 200 mM NaCl; 0.5% Triton X-100, TBS/T) and then incubated overnight at 4 °C with primary antibodies described in Appendix A. After washing with TBS/T, membranes were incubated with the horseradish-peroxidase-conjugated secondary antibody for 1 h at room temperature, followed by signal detection using enhanced chemiluminescence (ECL) (Amersham Bioscience, UK). The number of proteins was quantified using Image J software (National Institutes of Health, Bethesda, MD, USA).

### 2.7. TUNEL Assay

Apoptosis was examined in paraffin sections by the TUNEL-assay using DeadEnd™ Fluorometric TUNEL System (#G3250; Promega Corp., Madison, WI, USA) following the manufacturer’s protocol with modifications. Before the incubation with TdT enzyme and nucleotide mix for 1 h at 37 °C, sections were blocked with 5% BSA and normal goat serum diluted 1:5 in PBS before the addition of PNA lectin to mark the acrosome. Finally, the cell nuclei were counterstained with Vectashield + DAPI. The sections were observed and captured with an optical microscope (Leica DM 5000 B + CTR 5000) with a UV lamp and saved with IM 1000 software. To determine the % of TUNEL-positive cells, 30 seminiferous tubules/animal (N = 8) for a total of 240 tubules per group were counted.

### 2.8. Statistical Analysis

The values were compared by a Student’s *t*-test for between-group comparisons. Values for *p* < 0.05 were considered statistically significant. All data were expressed as the mean ± standard deviation (S.D.).

## 3. Results

### 3.1. The Level of D-Aspartate Strongly Decreased in the Testes of R26^Ddo/Ddo^ Mice

To assess whether the overexpression of DDO enzyme in *Ddo* knockin mice affected the endogenous levels of D-Asp in the testes, we performed HPLC in homozygous *R26^Ddo^*^/*Ddo*^ animals and their wild-type (*R26^+^*^/*+*^) littermates (Appendix A). We found a significant decrease in D-Asp levels in the testes of *R26^Ddo^*^/*Ddo*^ mice compared to that in the *R26^+^*^/*+*^ controls (Figure 1A). This finding was in line with our previous study on brain and peripheral tissues [37]. Then, we assessed the testicular levels of the D-Asp precursor, L-Asp, in *R26^Ddo^*^/*Ddo*^ and *R26^+^*^/*+*^ mice. We did not find any genotype-dependent variation of L-Asp between *R26^Ddo^*^/*Ddo*^ and *R26^+^*^/*+*^ mice (Figure 1B), which indicated the selective catabolic activity of DDO for the D-enantiomer of Asp [45]. Based on the selective changes in testicular D-Asp levels, our analysis also showed a significant reduction in the D-Asp/total Asp (D-Asp + L-Asp) ratio in *R26^Ddo^*^/*Ddo*^ mice, compared to the ratio in the *R26^+^*^/*+*^ littermates (Figure 1C).

### 3.2. The Lack of D-Aspartate Produced Morphological Abnormalities in the Testes of R26^Ddo/Ddo^ Mice

We conducted testicular histology in *R26^Ddo^*^/*Ddo*^ and *R26^+^*^/*+*^ mice (Figure 2A) to evaluate whether D-Asp depletion affected the morphology of the testes. The *R26^+^*^/*+*^ mice had a typical seminiferous epithelium (SE), presenting GC in all the stages of differentiation, tubular lumina with mature SPZ, and a typical interstitial compartment with LC and normal blood vessels. In contrast, the testes from the *R26^Ddo^*^/*Ddo*^ mice showed many empty spaces between GC and few SPZ in the lumina. The morphometric analysis showed differences in the histological architecture between the *R26^Ddo^*^/*Ddo*^ and *R26^+^*^/*+*^ testes (Figure 2B–D). The diameter of the tubules (Figure 2B; *p* < 0.01) and the thickness of the epithelium (Figure 2C; *p* < 0.001) were significantly lower in the testis of *R26^Ddo^*^/*Ddo*^ mice compared to their respective values in the testis of wild-type mice. The percentage of empty SPZ lumina was significantly higher in the testes of *R26^Ddo^*^/*Ddo*^ mice compared to those in the testes of *R26^+^*^/*+*^ controls (Figure 2D; *p* < 0.01).

### 3.3. The Lack of D-Aspartate Affected Steroidogenesis and Spermatogenesis in R26^Ddo/Ddo^ Mice

We first measured testosterone levels in the serum and testis of *R26^Ddo^*^/*Ddo*^ and *R26^+^*^/*+*^ mice and found significantly lower levels of testosterone in both serum (Figure 3A) and testis (Figure 3B) of *R26^Ddo^*^/*Ddo*^ mice compared to that in the testis of controls (*p* < 0.05).

We then analyzed the testicular levels of 17β-Hydroxysteroid dehydrogenase (17β-HSD), an enzyme that converts androstenedione into testosterone [2,46], via Western blotting assays. The expression of 17β-HSD in the testis of *R26^Ddo^*^/*Ddo*^ mice was significantly lower than that in the testis of the *R26^+^*^/*+*^ littermates (*p* < 0.05) (Figure 3C).

To evaluate whether the depletion of D-Asp affects the proliferative activity of GC, we investigated the expression levels of proliferating cell nuclear antigen (PCNA) and synaptonemal complex protein 3 (SYCP3) in the testis of *R26^Ddo^*^/*Ddo*^ and *R26^+^*^/*+*^ mice (Figure 3D,E). The results of the Western blotting analysis revealed that the levels of PCNA (Figure 3D) and SYCP3 (Figure 3E) proteins were significantly lower in the *R26^Ddo^*^/*Ddo*^ mice compared to that in the *R26^+^*^/*+*^ littermates (*p* < 0.05).

The effects of the lack of D-Asp on steroidogenesis and spermatogenesis were further confirmed by 17β-HSD and PCNA immunofluorescence (IF) staining (Figure 4). The 17β-HSD signal was specifically localized in the interstitial LC (asterisks), where it appeared weaker in *R26^Ddo^*^/*Ddo*^ mice than in *R26^+^*^/*+*^ controls (Figure 4A), as determined by the fluorescence intensity (*p* < 0.001; Figure 4B). Concerning PCNA, the results of IF staining showed its specific localization in the SPG layer at the basal tubules (striped arrows) and in the SPC (dotted arrows) in both genotypes (Figure 4A). However, the percentage of PCNA-positive cells was significantly lower in the testis of *R26^Ddo^*^/*Ddo*^ mice than in the testis of *R26^+^*^/*+*^ mice (*p* < 0.001; Figure 4C).

### 3.4. The Lack of D-Aspartate was Associated with an Increase in Testicular Apoptosis

We investigated the effect of the lack of D-Asp on testicular apoptosis by analyzing cytochrome c (Cyt c) protein levels in *R26^Ddo^*^/*Ddo*^ and *R26^+^*^/*+*^ mice. The expression of Cyt c in the testis of *R26^Ddo^*^/*Ddo*^ mice increased significantly compared to that in the testis of the *R26^+^*^/*+*^ littermates (*p* < 0.05) (Figure 5A,B). To confirm these findings, we performed a TUNEL assay. The results showed an increase in the number of positive cells by 63% (*p* < 0.01) (Figure 5C,D), particularly of SPG (striped arrows), in the *R26^Ddo^*^/*Ddo*^ mice compared to that in the *R26^+^*^/*+*^ controls.

### 3.5. The Lack of D-Asp Affected the Levels of Cytoskeleton Remodeling-Related Proteins

To further study the changes in the structural organization in the SE of *R26^Ddo^*^/*Ddo*^ testis, we analyzed the protein levels and localization of the proteins DAAM1 and PREP, which are involved in the cytoskeletal organization in the testis [35,36]. The results of the Western blotting analysis showed that DAAM1 protein level was lower in the testis of *R26^Ddo^*^/*Ddo*^ mice than in the testis of *R26^+^*^/*+*^ mice (*p* < 0.01; Figure 6A,B). Conversely, PREP protein level was higher in the testes of *R26^Ddo^*^/*Ddo*^ mice than in the testes of the *R26^+^*^/*+*^ littermates (*p* < 0.01; Figure 6A,C).

To examine the localization of DAAM1 and PREP in the testes of *R26^Ddo^*^/*Ddo*^ and *R26^+^*^/*+*^ mice, we performed a double IF staining of these proteins along with their respective cytoskeletal partner, i.e., actin and tubulin. As shown in Figure 6D, in the testis of *R26^+^*^/*+*^ mice, DAAM1 exhibited a quite diffused localization, considering that it was present in the perinuclear cytoplasm of SPG (striped arrows), SPC (dotted arrows), and SPT (arrowheads; Figure 6D, a). The results also revealed a co-localization of DAAM1 with β-actin in the cytoplasm of GC, highlighted by the intermediate yellow-orange tint (Figure 6D, a). Positive staining was also found in the LC (asterisks; Figure 6D, a). Conversely, in the testis sections of *R26^Ddo^*^/*Ddo*^ mice, a drastic staining decrease in most GC was observed, and a weak signal was detected in the perinuclear zone of SPG and SPC (Figure 6D, b). The analysis of fluorescence intensity confirmed the decrease in the levels of the DAAM1 protein in *R26^Ddo^*^/*Ddo*^ mice compared to that in the *R26^+^*^/*+*^ littermates (*p* < 0.01; Figure 6E).

In *R26^+^*^/*+*^ mice, PREP was mainly localized in the cytoplasmic protrusions of SC (arrows), where it was co-localized with tubulin (highlighted by the intermediate yellow-orange tint) and also occurred in the cytoplasm of elongating SPT (arrowheads; Figure 6D, c). A strong signal was also observed in the LC of *R26^+^*^/*+*^ mice (Figure 6D, c). In the testicular sections of *R26^Ddo^*^/*Ddo*^ mice, PREP localization pattern was similar to that of the *R26^+^*^/*+*^ littermates (Figure 6D, d). However, the staining of the SC and LC in the testes of *R26^Ddo^*^/*Ddo*^ mice was more intense than that in the testes of *R26^+^*^/*+*^ mice, as confirmed by the analysis of the fluorescence intensity (*p* < 0.01; Figure 6F).

Besides being involved in microtubule-associated processes, PREP acts as a peptidase that hydrolyzes small peptides, including GnRH [33,34]. Based on this finding, we analyzed the levels of GnRH in the testes of *R26^Ddo^*^/*Ddo*^ and *R26^+^*^/*+*^ mice by Western blotting assays. Our results showed an increase in testicular GnRH protein levels in *R26^Ddo^*^/*Ddo*^ mice compared to their levels in *R26^+^*^/*+*^ controls (*p* < 0.01; Figure 7A,B). We also assessed the co-localization of GnRH with PREP through IF staining. We found that in *R26^+^*^/*+*^ mice, GnRH was mainly localized in the interstitial compartment, and in the periphery of LC (asterisks; Figure 7C). We also confirmed the distribution pattern of PREP described above. In *R26^Ddo^*^/*Ddo*^ mice, GnRH and PREP retained the localization observed in the testes of *R26^+^*^/*+*^ controls (Figure 7C). However, the GnRH fluorescence intensity increased significantly compared to that in the *R26^+^*^/*+*^ mice (*p* < 0.01; Figure 7D). In the testes of *R26^Ddo^*^/*Ddo*^ and *R26^+^*^/*+*^ mice, GnRH and PREP co-localized in the cytoplasm of LC, as highlighted by the intermediate yellow-orange tint (Figure 7C).

## 4. Discussion

In this study, we used the *Ddo* knockin mouse model, characterized by the constitutive overexpression of DDO and the consequent depletion of D-Asp levels [37], to elucidate the role of this atypical amino acid in testicular function. Consistent with the findings of previous studies on the brain and other peripheral organs of *R26^Ddo^*^/*Ddo*^ [37], the results of our HPLC analysis indicated that one-month-old *Ddo* knockin mice showed almost complete depletion of D-Asp content also in the testis. Conversely, the amount of L-Asp was not affected, which supported the selectivity and specificity of DDO for D-Asp as the substrate.

Several studies have shown a strong correlation between D-Asp concentration and testosterone in rat testis [12,13]. The synthesis of D-Asp and testosterone starts in the testes in the late fetal stage, and their levels gradually increase after birth, reaching the highest levels when they are approximately 80 days old, i.e., upon reaching sexual maturity [10,12]. Similarly, Tomita et al. [7] showed that in the mouse testis, D-Asp levels are initially low and gradually increase to reach a maximum level at about 10 weeks of age. Several studies on adult mammals have shown that acute or chronic supplementation with D-Asp increases serum testosterone levels either through activation of the hypothalamus–pituitary–testis axis [12,47] or by directly upregulating the expression of steroidogenesis enzymes in the LC [2,17,18,19]. Based on these findings and considering that testosterone is critical for spermatogenesis, we investigated the effects of a dramatic reduction in D-Asp levels on the synthesis of testosterone in one-month-old *R26^Ddo^*^/*Ddo*^*mice*. Our results showed a significant decrease in both serum and testis testosterone levels in mice with testicular D-Asp depletion. We also found that the decrease in testosterone levels was coupled with a decrease in the expression of 17β-HSD protein (the enzyme catalyzing the conversion of androstenedione to testosterone). These findings confirmed our previous results demonstrating that D-Asp affects testosterone levels by regulating the expression of 17β-HSD protein [2].

Some studies have also shown that D-Asp can promote spermatogenesis by directly activating mitosis in SPG [21,36] and meiosis in SPC [23]. Specifically, in vivo and in vitro experiments have shown that D-Asp induces an increase in testicular protein expression of two mitotic markers, PCNA and p-H3, and the expression of SYCP3, which participates in meiosis [21,23,36]. In this study, we found that the expression levels of PCNA and SYCP3 were lower in the testis of *R26^Ddo^*^/*Ddo*^ mice, compared to that in the testis of the control littermates, suggesting that D-Asp depletion induces impaired spermatogenesis. This hypothesis was supported by histological and morphometric analyses in the *R26^Ddo^*^/*Ddo*^ mice, which indicated a general disorganization of the SE, with many empty spaces between GC and a significant decrease in tubular diameter and the thickness of the germinal epithelium. The decrease in spermatogenesis was further shown by the increase in the percentage of empty lumina in the testis of *R26^Ddo^*^/*Ddo*^ mice. In line with an overall influence of D-Asp on cell proliferation processes, our previous findings in newborn *R26^Ddo^*^/*Ddo*^ mice showed that D-Asp deficiency significantly decreased the number of proliferative neurons during brain corticogenesis, reflecting in the decrease in the corticostriatal gray matter volume in adulthood [48]. In contrast to peripheral glands, D-Asp levels are the highest in the brain during the embryonic and early post-natal phases and decrease sharply in adulthood [6,49]. Therefore, the period of the maximal emergence of D-Asp in the testes and brain matches very closely with the critical phases of their functional (testes) and morphological (brain) maturation, which are probably regulated by this D-amino acid.

Along with morphological abnormalities, we also found an increase in the percentage of TUNEL-positive cells in the testes of *R26^Ddo^*^/*Ddo*^ mice, which indicated an increase in the apoptosis of the seminiferous tubules in the absence of D-Asp. The TUNEL data were supported by biochemical results, which showed higher levels of the cytosolic Cyt c protein (a small hemeprotein released by the mitochondria when the intrinsic apoptotic pathway is activated) in the testes of *R26^Ddo^*^/*Ddo*^ mice. Mitochondria follow a peculiar pattern of activation during the maturation of male GC [50,51], and D-Asp stimulates mitochondrial dynamics and oxidative phosphorylation in GC2 cells [23]. Overall, our findings suggested that the increase in apoptosis in the testis of *R26^Ddo^*^/*Ddo*^ mice depended on the lack of D-Asp that could damage the mitochondrial compartment directly and indirectly through the reduction of testosterone levels.

In previous studies, we found that D-Asp promotes spermatogenesis by also increasing the expression of testicular DAAM1 [36] and PREP [35] which are essential for male fertility. Specifically, DAAM1 plays a dual role in type B SPG: (1) in the cytoplasm, it regulates actin remodeling during the differentiation phases of the seminiferous cycle; (2) in the nucleus, it allows actin polymerization, which is important for DNA replication and, eventually, cell division [36]. In this study, we found that the protein level of DAAM1 in the testis of the *R26^Ddo^*^/*Ddo*^ mice was lower than that in the testis of wild-type mice. Supporting the biochemical data, the results of the IF analysis showed a significant decrease in DAAM1 signal in most GCs. Therefore, a decrease in the expression of DAAM1 triggered by the lack of D-Asp might contribute to the impaired spermatogenesis observed in the testes of *R26^Ddo^*^/*Ddo*^ mice.

The results of the Western blotting analysis also indicated a higher protein expression of testicular PREP in *R26^Ddo^*^/*Ddo*^ mice, which was further corroborated by the IF data. These results showed a stronger PREP signal in the testis of *R26^Ddo^*^/*Ddo*^ mice, not only in the germinal compartment but also in the somatic components, such as SC and LC. The immunolocalization of PREP was particularly pronounced in the SC cytoplasmic protrusions, which are important for proper microtubule-based translocation of GC through the SE. Previous studies have shown that PREP is involved in microtubule-associated processes, including cytoskeletal remodeling, which occurs during GC movement [25]; these findings are consistent with the localization of PREP. Therefore, the enhanced PREP immunofluorescent signal detected in these cells of *R26^Ddo^*^/*Ddo*^ mice might be a mechanism to counteract the disorganization of the microtubule network, which could be one of the causes of the loss of GC. These results validated the hypothesis that the lack of D-Asp in the testis could alter the cytoskeleton of GC and SC, leading to the disruption of cell-cell junctions.

Besides its involvement in microtubule-associated processes, PREP acts as a peptidase and hydrolyzes small peptides, including GnRH [33,34]. More in detail, PREP participates in hormonal homeostasis, cleaving the C-terminal glycinamide residue of GnRH, thus limiting the stimulation of the pituitary gonadotropins, as demonstrated in the rat and ovine hypothalamic extracts [33,34]. We also found that GnRH and PREP were co-localized in the LC cytoplasm in both *R26^Ddo^*^/*Ddo*^ and wild-type mice. Our results indicated an increase in the fluorescence intensity of GnRH in the testes of *Ddo* knockin mice, which was also confirmed by the results of the Western blotting analysis. Thus, we hypothesized that the enhanced testicular expression of PREP in *R26^Ddo^*^/*Ddo*^, particularly in the LC, is also necessary for limiting the level of GnRH, which probably increases as a negative feedback response induced by the reduction in testosterone under conditions of D-Asp deficiency. In line with this assumption, in a previous study [35], we hypothesized that the D-Asp-induced increase in the expression of PREP in the LC is probably necessary to counteract and limit the level of GnRH, which is known to increase after D-Asp treatment [47]. In this regard, Venditti et al. [44] showed that an increase in PREP expression in adult rat testes, induced by maternal cadmium exposure during gestation and lactation, might regulate physiological sex hormone production/action. Our results also supported the hypothesis that PREP plays a key role in reproduction, considering that its testicular expression and localization are modulated in response to several endogenous and exogenous factors.

In conclusion, the *Ddo* knockin mouse is a valuable animal model for studying the role of D-Asp in steroidogenesis and spermatogenesis. In this study, we found that early D-Asp deficiency impairs testosterone biosynthesis in young mice, in which this hormone has not yet reached the maximum levels. Such changes can alter the proliferation and differentiation of GC.

## Figures and Tables

**Figure 1 biomolecules-13-00621-f001:**
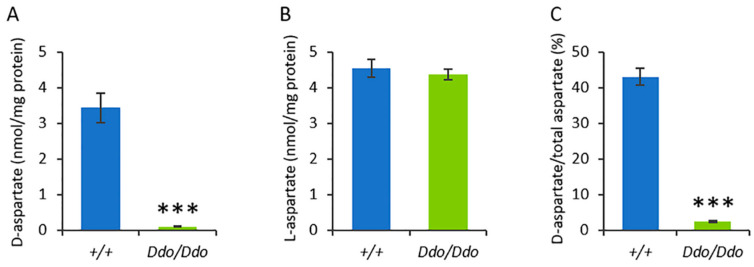
D-aspartate content strongly decreases in the testes of Ddo knockin mice. Detection of free D-aspartate (**A**) and L-aspartate levels (**B**), and D-aspartate/total aspartate ratio (**C**) in the testes of *R26^+^*^/*+*^ and *R26^Ddo^*^/*Ddo*^ mice (*n* = 3/genotype). *** *p* < 0.01, compared with *R26^+^*^/*+*^ mice (Student’s *t*-test). Values are expressed as the mean ± S.D. The amino acids were detected in a single run by HPLC and expressed as nmol/mg protein, while the ratio is expressed as a percentage (%).

**Figure 2 biomolecules-13-00621-f002:**
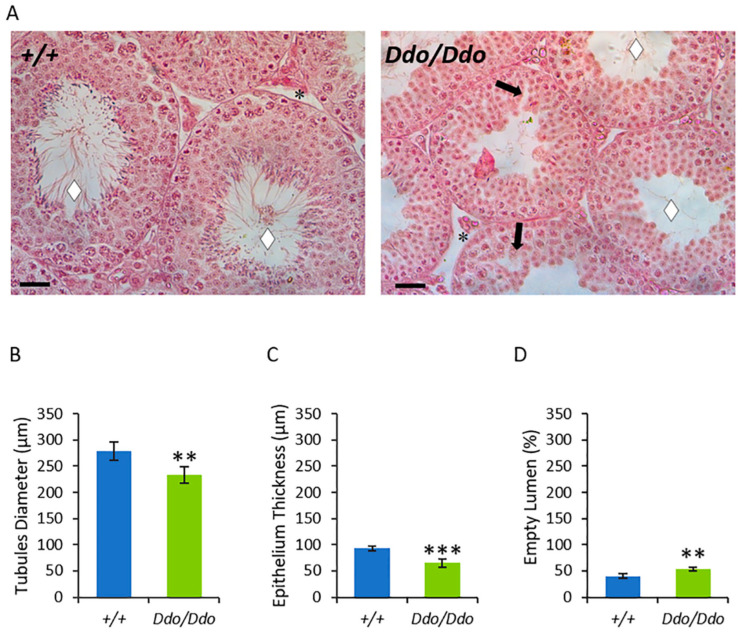
D-aspartate lack produces morphological abnormalities in the testes of Ddo knockin mice. (**A**) Paraffin sections of testes from *R26^+^*^/*+*^ and *R26^Ddo^*^/*Ddo*^ mice. Rhombus: tubules lumen; arrow: space between germ cells; asterisks: Leydig cells. Hematoxylin-eosin staining. Scale bars represent 40 µm. (**B**–**D**) Morphometric parameters in *R26^+^*^/*+*^ and *R26^Ddo^*^/*Ddo*^ testes were shown. The values represent the means ± S.D. of the values obtained from eight mice per genotype. ** *p* < 0.01; *** *p* < 0.001, compared with *R26^+^*^/*+*^ mice (Student’s *t*-test).

**Figure 3 biomolecules-13-00621-f003:**
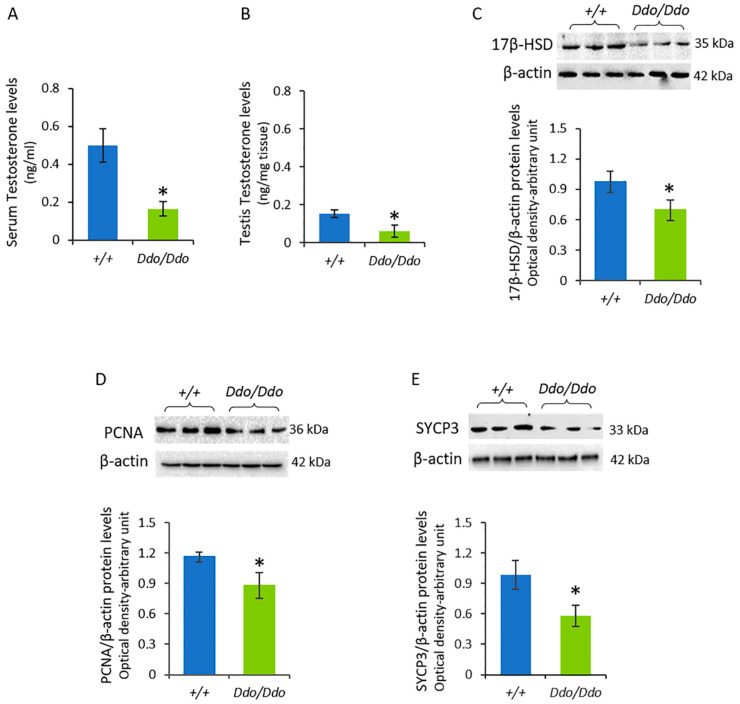
D-aspartate lack affects the levels of testosterone and proteins involved in steroidogenesis and spermatogenesis in the testes of Ddo knockin mice. Serum (**A**) and testis (**B**) testosterone levels, and Western blotting detections of 17β-HSD (**C**), PCNA (**D**), and SYCP3 (**E**) protein levels in the testes of *R26^+^*^/*+*^ and *R26^Ddo^*^/*Ddo*^ mice. A specific band of 35, 36, and 33 kDa, respectively, was detected. The protein levels were quantified using the ImageJ program and normalized with respect to β-actin protein (42 kDa). Representative immunoblots were shown above each panel. Data represent the means ± S.D. of the values obtained from eight mice per genotype. * *p* < 0.05, compared with *R26^+^*^/*+*^ mice (Student’s *t*-test).

**Figure 4 biomolecules-13-00621-f004:**
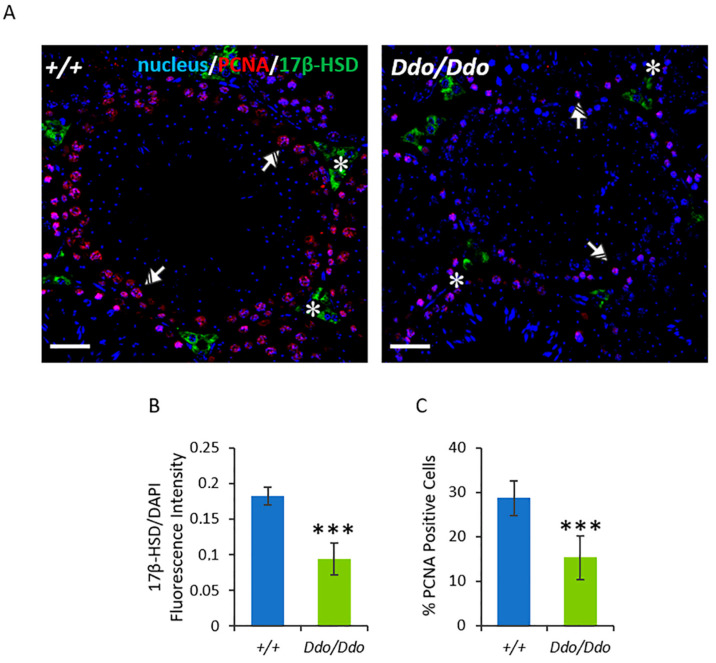
D-aspartate lack affects the levels and localization of proteins involved in steroidogenesis and spermatogenesis in the testes of Ddo knockin mice. 17β-HSD and PCNA localization in the testes of *R26^+^*^/*+*^ and *R26^Ddo^*^/*Ddo*^ mice. (**A**) Immunofluorescence analysis on 17β-HSD (green), PCNA (red), and nucleus (blue) in testes of *R26^+^*^/*+*^ and *R26^Ddo^*^/*Ddo*^ mice. Striped Arrows: spermatogonia; Dotted Arrows: spermatocytes; Asterisks: Leydig cells. Scale bars represent 20 μm. (**B**) Histogram showing the quantification of 17β-HSD fluorescence signal intensity. (**C**) Histogram showing the percentage (%) of PCNA-positive cells. All the values are expressed as means ± SD from eight mice per genotype. *** *p* < 0.001, compared with *R26^+^*^/*+*^ mice (Student’s *t*-test).

**Figure 5 biomolecules-13-00621-f005:**
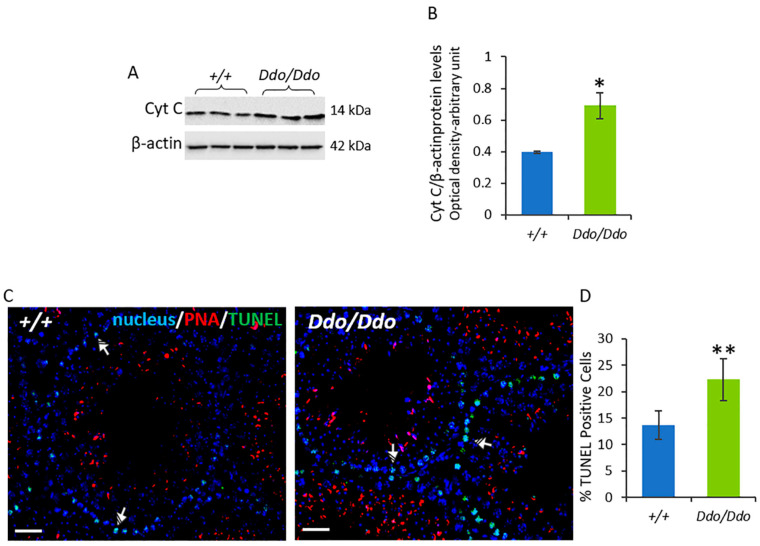
D-aspartate lack is associated with increased testicular apoptosis in Ddo knockin mice. *(***A**) Representative Western blotting detections for cytochrome C (Cyt C) protein levels in the testes of *R26^+^*^/*+*^ and *R26^Ddo^*^/*Ddo*^ mice. A specific band of 14 kDa was detected. (**B**) The protein levels were quantified using the ImageJ program and normalized with respect to β-actin protein (42 kDa). (**C**) Determination of apoptotic cells through the detection of TUNEL-positive cells (green) in the testes of *R26^+^*^/*+*^ and *R26^Ddo^*^/*Ddo*^ mice. Slides were counterstained with DAPI-fluorescent nuclear staining (blue) and with PNA lectin (red), which marks the acrosome. Scale bars represent 20 μm. Striped Arrows: spermatogonia. (**D**) Histogram showing the percentage (%) of TUNEL-positive cells. The data represent the means ± S.D. of the values obtained from eight mice per genotype. * *p* < 0.05, ** *p* < 0.01, compared with *R26^+^*^/*+*^ mice (Student’s *t*-test).

**Figure 6 biomolecules-13-00621-f006:**
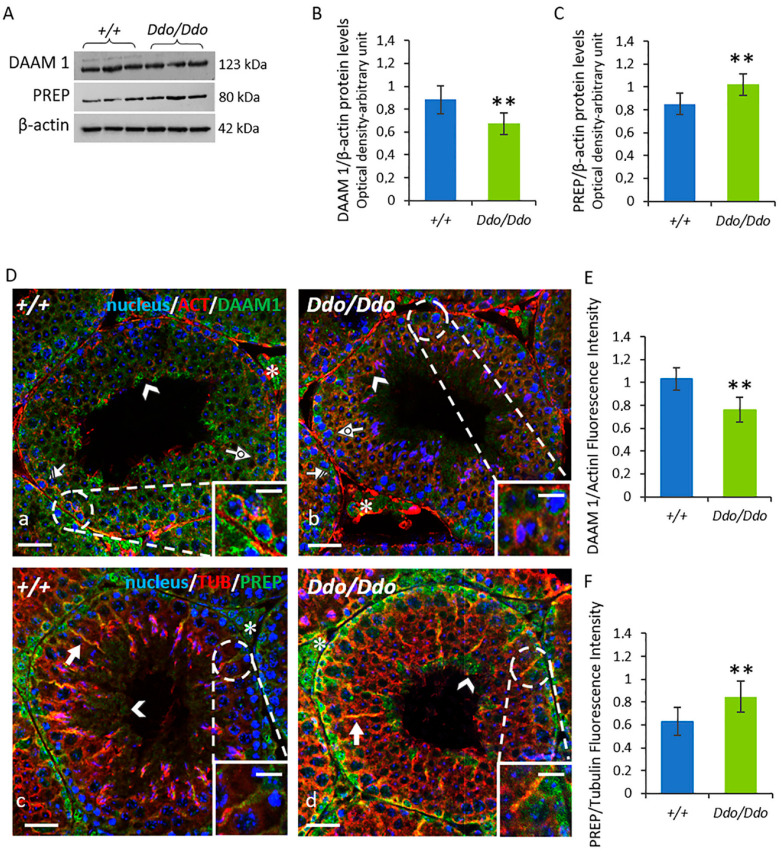
D-aspartate lack affects the levels of cytoskeleton remodeling-related proteins in Ddo knockin mouse testes. (**A**) Representative Western blotting detections of DAAM1 (123 kDa), PREP (80 kDa), and β-actin (42 kDa) protein levels in testes of *R26^+^*^/*+*^ and *R26^Ddo^*^/*Ddo*^ mice. (**B**,**C**) The protein levels of DAAM1 and PREP were quantified using the ImageJ program and normalized with respect to β-actin protein (42 kDa). (**D**) Immunofluorescence analysis of DAAM1 in testes of *R26^+^*^/*+*^ (**a**) and *R26^Ddo^*^/*Ddo*^ (**b**) mice: DAAM1 (green), β-actin (red), and nucleus (blue); Immunofluorescence analysis of PREP in testes of *R26^+^*^/*+*^ (**c**) and *R26^Ddo^*^/*Ddo*^ (**d**) mice: PREP (green), α-tubulin (red) and nucleus (blue). Striped Arrows: spermatogonia; Dotted Arrows: spermatocytes; Arrowheads: elongating spermatids; Arrows: Sertoli cells; Asterisks: Leydig cells. Scale bars represent 20 μm and 10 μm in the insets. (**E**,**F**) Histograms showing the quantification of DAAM1 and PREP fluorescence signal intensity, respectively. Data represent the means ± S.D. of the values obtained from eight mice per genotype. ** *p* < 0.01, compared with *R26^+^*^/*+*^ mice (Student’s *t*-test).

**Figure 7 biomolecules-13-00621-f007:**
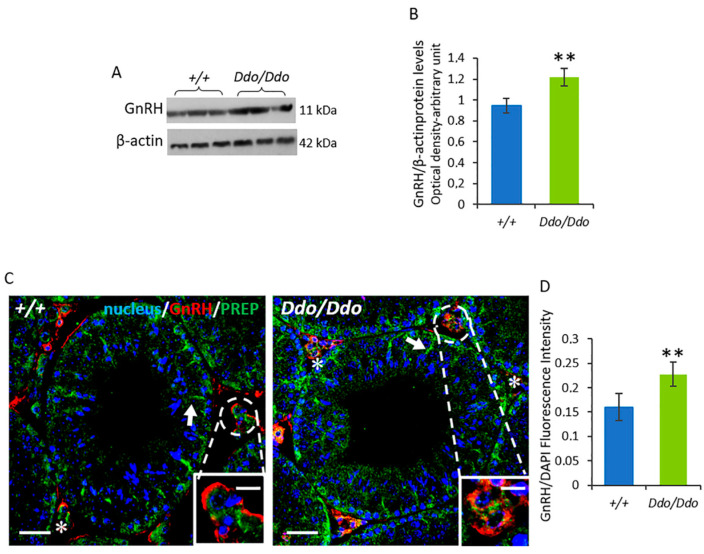
D-aspartate lack affects GnRH protein levels and localization in Ddo knockin mouse testes. (**A**) Western blotting analysis showing the protein levels of GnRH (11 kDa) and β-actin (42 kDa) in testes of *R26^+^*^/*+*^ and *R26^Ddo^*^/*Ddo*^ mice. (**B**) GnRH protein level was quantified using the ImageJ program and normalized with respect to β-actin protein (42 kDa). (**C**) Immunofluorescence analysis of GnRH (red), PREP (green), and nucleus (blue) in testes of *R26^+^*^/*+*^ and *R26^Ddo^*^/*Ddo*^ mice. Arrows: Sertoli cells; Asterisk: Leydig cells. Scale bars represent 20 μm and 10 μm in the insets. (**D**) Histogram showing the quantification of GnRH fluorescence signal intensity. Data are expressed as means ± SD from eight mice per genotype. ** *p* < 0.01, compared with *R26^+^*^/*+*^ mice (Student’s *t*-test).

## Data Availability

The data presented in this study are available on request from the corresponding author.

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
