# Peer review of "D-Aspartate Depletion Perturbs Steroidogenesis and Spermatogenesis in Mice"

_biomolecules, 2023, doi:10.3390/biom13040621_

Round 1

Reviewer 1 Report

Thanks to the editor for this kind invitation, and I have reviewed this manuscript biomolecules-2266297carefully.

There have been many studies on the role of D-aspartate in spermatogenesis. Therefore, in terms of innovation, the manuscript is not novel enough.

In terms of the overall content, the author's display of the phenomenons is greater than the exploration of the mechanism, and the depth of the article is not enough.

To be sure, the immunofluorescence picture has been done quite well, and if the authors can further explore the mechanism, the quality of the article may be greatly improved.

Yours Sincerely

Author Response

We thank the Reviewer for the comments, but we would like to point out the differences between this paper and the previous ones concerning the role of D-Asp in spermatogenesis (Di Fiore et al., 2019 for review). Our group has been studying the effects of the exogenous D-Asp on reproduction for a long time, and the novelty of this paper is to use, for the first time, a Ddo knockin mouse model, in which constitutive overexpression of DDO produces a complete depletion of D-Asp levels. This mouse model allows the study of testicular activity in the absence of D-Asp and is consequently of interest and helpful in highlighting the "problems" resulting from the lack of D-Asp.

In this paper, we assessed that testosterone levels are low, as is the protein level of a steroidogenic enzyme (17b-HSD). These observations may account for the increased germ cell apoptosis and reduced proliferation. In addition, altered spermatogenesis is a consequence of impaired expression and localization of cytoskeleton-associated proteins. For these reasons, the present work should be considered as the necessary starting point to verify the events resulting from the lack of D-Asp and then to perform further experiments exploring the basal mechanisms induced by D-Asp on spermatogenesis.

Reviewer 2 Report

The manuscript is written readably focusing on the role of D-Asp on testicular function and its possible mechanism. The experimental design and scientific integrity of this research are quite appreciable. However, I believe it has a few minor problems that can be improved. The manuscript will be accepted upon revision. The following are more detailed comments.

Please better describe testosterone assays-it is not clear.

Some P values in the manuscript do not have italics-please revise them.

Author Response

The manuscript is written readably focusing on the role of D-Asp on testicular function and its possible mechanism. The experimental design and scientific integrity of this research are quite appreciable. However, I believe it has a few minor problems that can be improved. The manuscript will be accepted upon revision. The following are more detailed comments.

We thank the Reviewer for appreciating our manuscript.

Please better describe testosterone assays-it is not clear.

We better described Testosterone assay in the Materials and Methods section.

Some P values in the manuscript do not have italics-please revise them.

The revision has been performed accordingly.

Reviewer 3 Report

In order to consider publication, the authors must respond to the comments submitted.

The work is well planned, and the experiments have been correctly designed to answer the suggested hypothesis. However, I consider that some issues should be clarified to ensure the understanding and reliability of the work.

Abstract:

Abbreviations must be specified the first time they are named:

disheveled-associated activator of morphogenesis 1 DAAM1

prolyl endopeptidase PREP

 Introduction:

Why are these two proteins used as markers? Could you provide a bibliography from other authors on the relevance of these two proteins for fertility?

L-69: proper cytoskeleton dynamic function

L-76: It is not clear if you are talking about D-Asp or PREP for readers here. Revise punctuation to improve understanding.

Materials and methods:

Ddo knockin mouse: The ethics committee that approves the procedure must be included. The experimentation guide used for the procedure must also be included (UNESCO, ARRIVE…).

l-123: National Institutes of Health (NIH) would be replaced by (National Institutes of Health, 156 Bethesda, USA).

L-165: optical microscopy replaces by an optical microscopy.

Results:

Figure 1- It is possible to see the chromatogram where the D-aspartate peak is observed? it would be advisable to attach it in supplementary material

Figure 2A: Put the scale bar inside the image.

Figure 2B: I think it would be easier to read if there were three separate graphs. It would not be necessary to repeat the concepts on the y and x axis.

Figure 3A. Put serum and testis testosterone levels on two different graphs because the units are different. Thus, the information on the X-axis would not have to be repeated. That would be easier to read.

3C/3D: SYCP3 and PCNA have been performed on the same membrane? The tubulin control used appears to be the same. Please clarify this point.

4A: Change the color of 17b-HSD to improve visualization.

4B: Fluorescence intensity units (arbitrary unit, quantification method…).

5C, 6D, and 7C: put the scale bar inside the image.  As a suggestion, the insets could be larger to improve visualization. 

Author Response

In order to consider publication, the authors must respond to the comments submitted.

The work is well planned, and the experiments have been correctly designed to answer the suggested hypothesis. However, I consider that some issues should be clarified to ensure the understanding and reliability of the work.

We greatly appreciated the Reviewer for the comments and valuable suggestions, which have allowed us to improve the article.

Abstract:

Abbreviations must be specified the first time they are named: disheveled-associated activator of morphogenesis 1 DAAM1, prolyl endopeptidase PREP

The meaning of DAAM1 and PREP acronyms have been specified.

 Introduction:

Why are these two proteins used as markers? Could you provide a bibliography from other authors on the relevance of these two proteins for fertility?

We thank the reviewer for the questions to clarify in the introduction the choice of PREP and DAAM1 as markers. However, we would like to point out that only a few other Authors have worked on PREP in spermatogenesis while regarding DAAM, we were the first to demonstrate the role of this protein in reproductive activity, but recently another group published work on DAAM2 in rat Sertoli cells (Xu et al., 2023).

L-69: proper cytoskeleton dynamic function

We corrected the sentence.

L-76: It is not clear if you are talking about D-Asp or PREP for readers here. Revise punctuation to improve understanding.

We corrected the sentence.

Materials and methods:

Ddo knockin mouse: The ethics committee that approves the procedure must be included. The experimentation guide used for the procedure must also be included (UNESCO, ARRIVE…).

We thank the Reviewer for pointing out the lack of this important disclosure that we missed in the previous version of the manuscript. In the new Materials and Methods section of the text, we have now included the ethics committee approval.

l-123: National Institutes of Health (NIH) would be replaced by (National Institutes of Health, 156 Bethesda, USA).

We corrected the sentence.

L-165: optical microscopy replaces by an optical microscopy.

We corrected the sentence.

Results:

Figure 1- It is possible to see the chromatogram where the D-aspartate peak is observed? it would be advisable to attach it in supplementary material

We thank the Reviewer for her/his appropriate request. We have now added, as a new supplementary figure, a representative HPLC chromatogram showing the specificity of D-aspartate peak in the testis of our mice.

Figure 2A: Put the scale bar inside the image.

The scale bar is already present inside the image.

Figure 2B: I think it would be easier to read if there were three separate graphs. It would not be necessary to repeat the concepts on the y and x axis.

The graphs have been divided accordingly.

Figure 3A. Put serum and testis testosterone levels on two different graphs because the units are different. Thus, the information on the X-axis would not have to be repeated. That would be easier to read.

The graphs have been divided accordingly.

3C/3D: SYCP3 and PCNA have been performed on the same membrane? The tubulin control used appears to be the same. Please clarify this point.

The Reviewer is correct, and we thank you for this comment; we made a mistake in preparing the figures. PCNA and SYCP3 were not analyzed on the same filter because 1) PCNA and SYCP3 have very similar molecular weights and 2) both anti-PCNA and anti-SYCP3 are raised in mice, so the secondary antibody might recognize the same signal if stripping is not done correctly. We have included the corrected image of beta-actin.

4A: Change the color of 17b-HSD to improve visualization.

The label in figure 4A have been changed, accordingly.

4B: Fluorescence intensity units (arbitrary unit, quantification method…).

The information has been added.

5C, 6D, and 7C: put the scale bar inside the image.  As a suggestion, the insets could be larger to improve visualization. 

The scale bars are already present inside the images. According to Reviewer’s suggestion, we have enlarged the insets.